

# Dynamic weighted ensemble of geoscientific models via automated machine learning-based classification

Hao Chen[1,2,4], Tiejun Wang[1,2,3], Yonggen Zhang[1,2], Yun Bai[5] and Xi Chen[1,2,3]

[1]Institute of Surface-Earth System Science, School of Earth System Science, Tianjin University, Tianjin, 300072, China

[2]Tianjin Key Laboratory of Earth Critical Zone Science and Sustainable Development in Bohai Rim, Tianjin University, Tianjin, 300072, China

[3]Tianjin Bohai Rim Coastal Earth Critical Zone National Observation and Research Station, Tianjin University, Tianjin, 300072, China

[4]State Key Laboratory of Remote Sensing Science, Aerospace Information Research Institute, Chinese Academy of Sciences, 10 Beijing, 100101, China

[5]Research Center for Remote Sensing Information and Digital Earth, College of Computer Science and Technology, Qingdao University, Qingdao, 266071, China

*Correspondence to*: Tiejun Wang (tiejun.wang@tju.edu.cn)

**Abstract.** Despite recent developments in geoscientific (e.g., physics/data-driven) models, effectively assembling multiple models for approaching a benchmark solution remains challenging in many sub-disciplines of geoscientific fields. Here, we proposed an automated machine learning-assisted ensemble framework (AutoML-Ens) that attempts to resolve this challenge. Details of the methodology and workflow of AutoML-Ens were provided, and a prototype model was realized with the key 20 strategy of mapping between the probabilities derived from the machine learning classifier and the dynamic weights assigned to the candidate ensemble members. Based on the newly proposed framework, its applications for two real-world examples (i.e., mapping global soil water retention parameters and estimating remotely sensed cropland evapotranspiration) were investigated and discussed. Results showed that compared to conventional ensemble approaches, AutoML-Ens was superior across the datasets (the training, testing, and overall datasets) and environmental gradients with improved performance 25 metrics (e.g., coefficient of determination, Kling-Gupta efficiency, and root mean squared error). The better performance suggested the great potential of AutoML-Ens for improving quantification and reducing uncertainty in estimates due to its two unique features, i.e., assigning dynamic weights for candidate models and taking full advantage of AutoML-assisted workflow. In addition to the representative results, we also discussed the interpretational aspects of the used framework and its possible extensions. More importantly, we emphasized the benefits of combining data-driven approaches with physics 30 constraints for geoscientific model ensemble problems with high dimensionality in space and non-linear behaviors in nature.



## 1 Introduction

With improvements to sensing systems and modeling technologies, a wide range of physics-based or data-driven models have been developed in the sub-fields of geosciences, mainly to simulate or predict essential variables for understanding climate, biodiversity, ocean, and geodiversity (Hurrell et al., 2013; Karpatne et al., 2019; Reichstein et al.,
2019). However, significant precision inconsistencies exist among these models due to their own limitations, even for the same process or variable on an identical scale (Steffen et al., 2020). It is, therefore, not surprising that the corresponding simulations or predictions are often different or even contradictory, particularly with the influence of anthropogenic activities in Earth systems, leading to the increasing need for better theories, methods, and data sets (Abbott et al., 2019; Tortell, 2020).

As a critical flux variable that links water, energy, and carbon cycling, a variety of terrestrial evapotranspiration (ET)
products are currently available at regional and global scales (Mueller et al., 2013), which are derived from various sources and/or approaches, including in-situ observations, land surface models, satellite inversion, and estimates from data-driven algorithms (Pan et al., 2020). Although these ET products provide an indispensable tool for investigating ET and its related processes (Han et al., 2020; Jung et al., 2010; Pascolini-Campbell et al., 2021), they often exhibit considerable discrepancies across diverse biomes and climate regimes, which could be attributed to a number of reasons, such as differences in model
structure and parameterization, input data, and scaling problems (Pan et al., 2020). In particular, no ET products with consistently low noise levels over time and space were found (Mueller et al., 2013), and therefore how to approach a benchmark ET data set remains a major challenge. To tackle this issue, it is advocated to apply model ensemble approaches to enhance the precision of available ET products (Lu et al., 2021), as previous studies have demonstrated the superiority of using ensemble strategies over any of the single models (Fragoso et al., 2018; Maclin and Opitz, 1999; Zounemat-Kermani
et al., 2021).

In this context, increasing efforts have been devoted to assembling multiple geoscientific models to improve quantification and reduce uncertainty in estimations (Araújo and New, 2007; Palmer et al., 2005; Reshmidevi et al., 2018). Numerous ensemble methods have been proposed, ranging from simple methods such as arithmetic mean (referred to as MEAN) to more complicated ones such as weighted mean using the Bayesian model averaging (BMA), empirical orthogonal
function (EOF), and reliability ensemble average (REA) approaches (Lu et al., 2021). For example, Dai et al. (2019a) reported a fitting method to obtain a global data set of hydraulic and thermal parameters of the soil from the ensemble pedotransfer functions (PTFs), which led to greater reliability than the median values of various PTFs (Dai et al., 2013). Chen et al. (2019a; 2019b) constructed a combined terrestrial water storage anomaly (TWSA) series by assigning time-dependent weights for five GRACE TWSA solutions, with the lowest noise level compared to other single solutions. Other
ensemble approaches have also been proposed, such as least-squares and maximizing temporal correlation techniques for merging soil moisture products (Kim et al., 2015; Yilmaz et al., 2012), conditional merging and geographic ratio analysis for precipitation data fusion (Duan and Bastiaanssen, 2013; Jongjin et al., 2016), and deep learning-based multi-dimensional ensemble methods for short-term runoff prediction (Liu et al., 2022). In general, those studies showed that the use of





ensemble approaches could virtually reduce the uncertainties of the data products by deriving and assigning their weights to generate the merged ones.

It should be noted that currently available ensemble approaches usually provide fixed weights to each candidate according to either their statistical degree of approximation to sparse observations or relative uncertainties without comparing to true variables (see, e.g., Fragoso et al., 2018; Liu et al., 2022; Madadgar et al., 2014; Tebaldi et al., 2005). Since environmental factors jointly and non-linearly regulate underlying processes, assigning fixed weights under all

conditions to individual models that depend on just a subset of constraints may not fully utilize the strength of ensemble approaches and/or individual models (Bai et al., 2021; Telteu et al., 2021). Therefore, it underscores the universality and importance of a particular issue, i.e., multiple models always exist while an effective ensemble one is still necessary towards better estimations (e.g., Abramowitz et al., 2019). To that end, it is still warranted to investigate and develop innovative methods based on ensemble model frameworks.

With increasing data availability for earth systems, machine learning (ML) techniques provide additional avenues for addressing this issue (e.g., Zounemat-Kermani et al., 2021). As an illustration, Zaherpour et al. (2019) proposed a unique application of ML to deliver optimized combinations of multiple global hydrological model (GHM) simulations, with considerably improved performance compared to the best performing GHM. Bai et al. (2021) presented four ensemble models based on ML to assemble six physics-based ET models to map cropland ET. Their ensembles can unify the

capabilities of various environmental constraints on ET utilized by specific models. However, the use of ML models is still faced with several challenges, such as feature engineering, model/optimization algorithm selection, and neural architecture design, making it time-consuming and error-prone if constructed manually (Tuggener et al., 2019).

In contrast, state-of-the-art automated ML (AutoML) appears to take the human factor out of these complex ML pipelines (Yao et al., 2018). Like ML approaches, AutoML is a computer program that has acceptable generalization

performance on input data and given tasks. The critical difference is that AutoML emphasizes the construction of high-level controlling approaches (i.e., what and how to automate) to use ML tools effectively and optimally, leading to new levels of capability and customization (Truong et al., 2019). For instance, Sun et al. (2021) applied an AutoML workflow (comprising six types of ML algorithms and various sets of predictors) to perform gridded water storage reconstruction over the conterminous United States (CONUS). The authors found that no one ML algorithm could reach the best reconstruction

performance across the CONUS, underscoring the importance of adopting an AutoML workflow to train, improve, and merge different ML methods to achieve robust performance. Nowadays, a host of AutoML tools and platforms, both free/open-source and commercially available, have been released for various scientific and engineering applications, e.g., Auto-Weka, TPOT, AutoKeras, Auto-Sklearn, H2O-Automl, Google Cloud Automl, and Microsoft AzureML (see the review by Truong et al. (2019). However, a comprehensive comparison among these different platforms to solve given

problems is another crucial issue beyond the scope of this study.

Based on the above discussions, the objectives of this study were to 1) introduce an AutoML-based ensemble (AutoML-Ens) framework for assembling multiple geoscientific models, and 2) present examples with the proposed AutoML-Ens



framework, including mapping global soil water retention parameters and improving remote sensing-based cropland ET estimates. In the following, Section 2 covers the details of the methodology and workflow of the AutoML-Ens framework, 100 and Section 3 presents data acquisition, results, and discussion about the two representative applications, followed by conclusions in Section 4.

## 2 Proposed AutoML-Ens framework

### 2.1 Methodology and workflow of AutoML-Ens

The overall pipeline of the proposed AutoML-Ens framework is illustrated in Figure 1. The main strategy of AutoML- 105 Ens is based on varying weights, i.e., weights assigned to candidate ensemble members vary depending on the spatial and temporal changes in environmental conditions and the performance capabilities of individual models under these conditions. Specifically, once a multimodel ensemble problem is defined, an extensive spectrum of meaningful predictors (i.e., environmental conditions) denoted by $x_m$, where $m = 1, \cdots, M$ with a single or a combination of few subsets are selected and used to develop physics-constrained models (hereafter the predictions $P_s$ where $s = 1, \cdots, S$).

$$P_s = f(x_m, \cdots) \tag{1}$$

where $x$ is the vector representing a predictor that can be a static or spatiotemporal-varying environmental variable; the vector $P$ denotes the predictions of different models; and the subscripts $m$ and $s$ represent the index of a predictor and model, respectively.

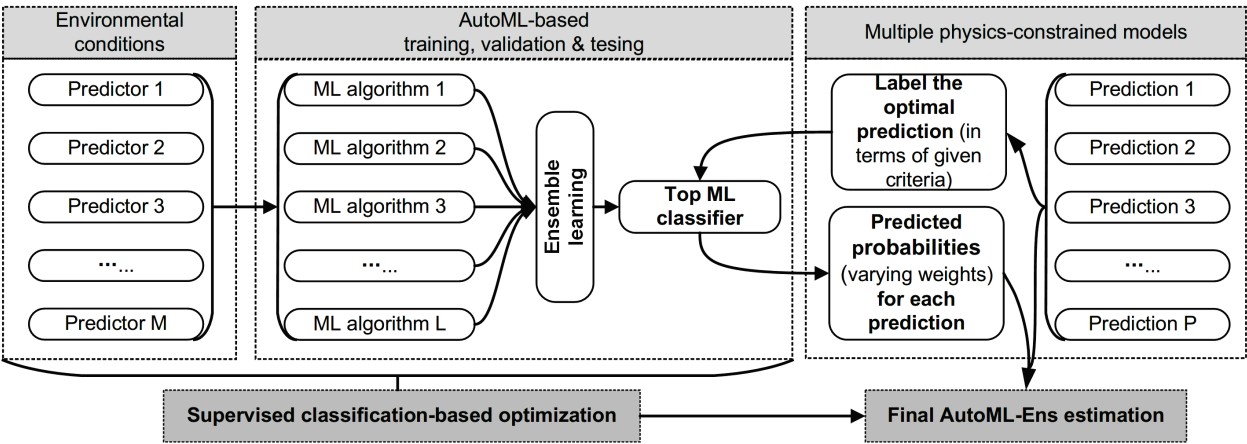

**Figure 1.** Procedures for building an AutoML-based ensemble framework (AutoML-Ens) to assemble geoscientific models.

To determine the ideal weights ($W_k$) for various models ($P_k$), we use an ML classifier to calculate the probability (designated as $W_k$) that each model is optimal in a certain environmental state. Especially, the ML classifier is trained to



find the optimal models labeled as those that produce predictions with specific criteria (e.g., the least absolute error compared against observations) under a specific environmental condition. Thus, ML classifiers can approximate model
weights with only factors that reflect the environment after training. Here, an AutoML-based training, validation, and testing workflow is conducted to help automatically find the top classifier $C_T$ (either a specific ML algorithm or an ensemble of a few ML algorithms $[M_l, l = 1, \cdots, L]$, based on the ensemble learning technique). The final AutoML-Ens estimation ($Y$) can subsequently be obtained by combining these candidate predictions ($P$) and their corresponding probabilities (i.e., varying weights $W$) derived by the AutoML-based $C_T$.

$$Y = [y_1, y_2, \cdots, y_K] \tag{2}$$

$$y_k = P_k \times W_k \tag{3}$$

$$P_k = [p_{k,1}, p_{k,2}, \cdots, p_{k,N}] \tag{4}$$

$$W_k = [w_{k,1}, w_{k,2}, \cdots, w_{k,N}]^T \tag{5}$$

where the vector $Y$ represents the final AutoML-Ens estimation; the subscript $k$ refers to the sample index of a model
prediction that can be spatially and/or temporally varying, thus $y_k$ denotes the ensemble of multimodel predictions for the sample $k$; $W_k$ is the varying weights associated with the multiple predictions $P_k$ for sample $k$. These weights are derived from an AutoML-based classifier, that is, the probability of an individual model being optimal under certain environmental conditions, and $\sum_{n=1}^{N} w_{k,n} = 1$; the subscript $K$ and $N$ are the numbers of samples and models, respectively.

Accordingly, two distinguishing features of AutoML-Ens can be stated as follows: 1) it focuses on assembling multiple
physics-constrained models to seek the optimal combination of physical and data-driven solutions, and 2) it is a supervised classification-based optimization that realizes the mapping between ML classifier-derived probabilities and dynamic adaptivity (or weights) used for an ensemble estimation to capture the non-linear nature of targeted processes and takes full advantage of AutoML-assisted workflow. In addition, it is noteworthy that most AutoML platforms support both a collection of existing ML algorithms to select the best one and their ensembles (referred to as the pure AutoML-based ensemble, P-
AutoML-Ens) based on 'ensemble learning' (see Figure 1) techniques such as bagging, boosting, dagging, and stacking (Zounemat-Kermani et al., 2021). Although both can be implemented on the AutoML platform, there are significant differences in the target ensemble objects and the strategies used between the proposed AutoML-Ens and these P-AutoML-Ens. Specifically, the core of the proposed AutoML-Ens is an ML classifier, and in order to obtain the optimal classifier, the inherent multiclassifier ensemble learning approaches in the AutoML platforms could be used. Meanwhile, for P-AutoML-
Ens, the 'ensemble' here is not aimed at assembling multiple models constrained by physics but the ML algorithms involved for given tasks. For example, we can select various ML algorithms to predict a target variable as a regression task without



physical constraints. The AutoML tools can then help to assemble these pure data-driven algorithms inherently to make the final better estimation. Further comparison and discussion of AutoML-Ens and P-AutoML-Ens can be found in Section 3.2.2.

## 2.2 A prototype AutoML-Ens for geoscientific examples

In this study, we built a prototype AutoML-Ens in the R environment (V3.6.3) using the H2O-AutoML platform (V3.32.1.7) in H2O.ai (Ledell and Poiri, 2020). H2O-AutoML is one of the leading open-source AutoML platforms according to recent benchmarking tests (Truong et al., 2019). The algorithms available in H2O-AutoML include some of the most commonly used ML algorithms and their variants, e.g., deep neural network (DNN), distributed random forest (DRF), generalized linear model (GLM), gradient boosting machine (GBM), extreme gradient boosting (XGBoost), and extremely
randomized trees (XRT). Furthermore, H2O-AutoML provides a stacking process to find the best combination of algorithms to obtain better predictive performance, which can be recognized as one kind of realization form of P-AutoML-Ens. Detailed explanations of H2O.ai and its H2O-AutoML platform can be found in Ledell and Poiri (2020). Here, the common features of AutoML-Ens for the examples are summarized below. 1) In the H2O-AutoML pipeline, the data (i.e., predictors and labels) are randomly shuffled into training (75% with five equal-sized subsets for cross-validation) and testing (25%). Note
that due to the use of the automatic hyperparameter optimization based on cartesian or random grid search methods in an H2O-AutoML run (Ledell and Poiri, 2020), the maximum number of ML models was set to be 30, in addition to the two ensemble models stacked (one with the highest performance model of each algorithm family and the other with all training models). Then, all 32 models were ranked to select the best ML classifier for final estimations. 2) Two widely used ensemble methods (that is, MEAN and BMA) were chosen for comparison (here, BMA was performed using the package
'EBMAforecast' (Montgomery et al., 2017) in the R environment). In addition, the hierarchical multimodel ensemble (HME) approach proposed by Zhang et al. (2020) to estimate soil water retention parameters, and the multilayer perception neural network classifier (MLP) introduced by Bai et al. (2021) with the most efficient in terms of accuracies and costs for assembling multiple physically driven cropland ET models, were also investigated as baseline models, respectively. An overview of the MEAN, BMA, HME, and MLP methods we used is presented in Supplementary Text S1. 3) Regarding the
performance evaluations for different models and/or ensembles, several statistical metrics, namely the Kling-Gupta efficiency (KGE), the coefficient of determination ($R^2$), and the root mean squared error (RMSE), were utilized.

## 3 Illustrative examples

    Two real-world examples are presented in this section to test the viability of using AutoML-Ens for tackling geoscientific model ensemble problems.





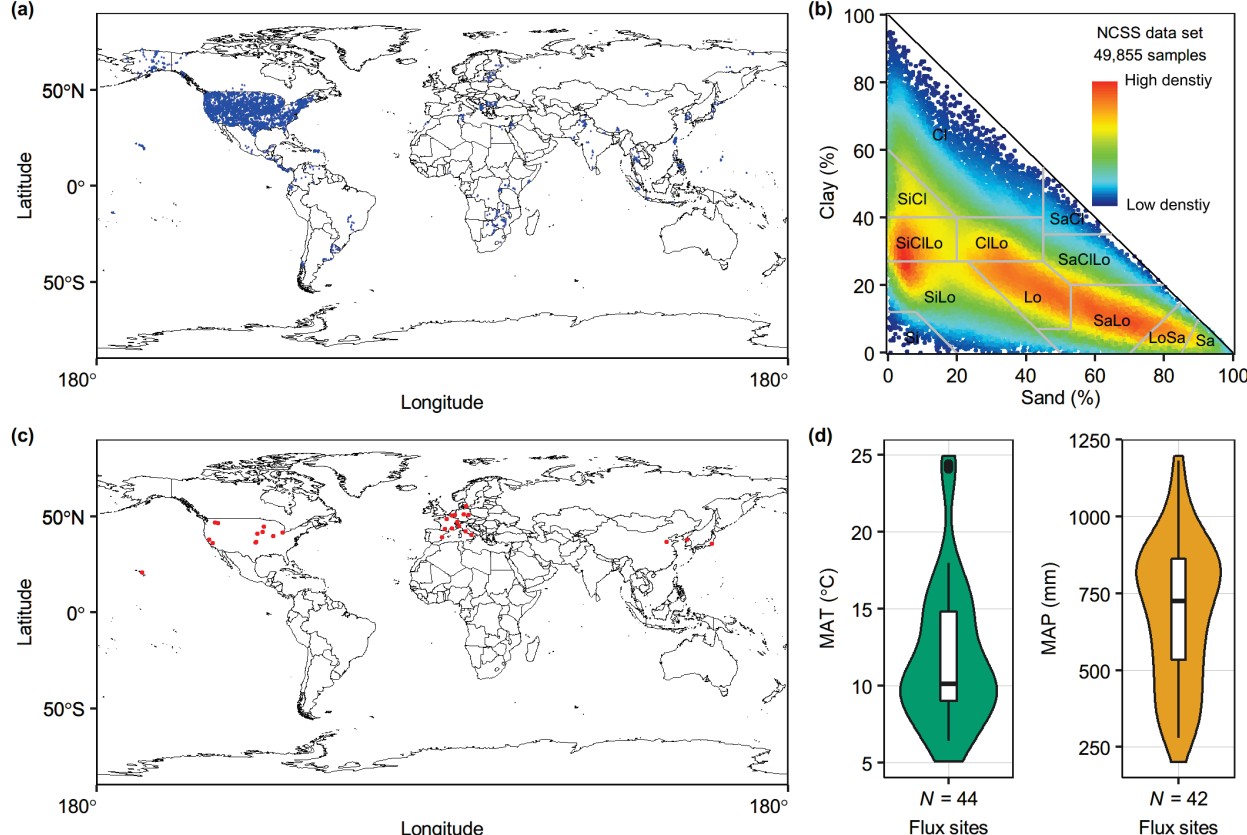

**Figure 2. (a)** Locations of selected soil samples from the National Cooperative Soil Survey Characterization (NCSS) covering the conterminous United States (87.7% of the data) and other regions of the globe (12.3% of the data) and their density distribution plotted in **(b)** the US Department of Agriculture soil textural triangle (USDA). **(c)** Locations of 47 eddy covariance flux sites that cover croplands from AmeriFlux, AsiaFlux, FLUXNET, and the European Flux Database Cluster, and **(d)** their mean annual temperature (MAT, °C) and mean annual precipitation (MAP, mm) distributions.

## 3.1 Mapping global soil water retention parameters

### 3.1.1 Related work and data acquisition

Accurate mapping of soil water retention characteristics is essential to quantify mass-energy exchanges between the terrestrial surface and the atmosphere but is challenged by limited measurements across the globe (Dai et al., 2019b). Empirical models (i.e., PTF) often use available soil attributes. (e.g., soil texture, bulk density-BD, and soil organic matter content-OC), have been developed to estimate soil hydraulic properties, e.g., hydraulic conductivity and water retention parameters (Van Looy et al., 2017). However, despite various advancements, the reliability of PTFs for global estimates is generally uncertain, given their nonlinearities and heterogeneities (Jena et al., 2021). Thus, the assembly of multiple PTFs





has been highly recommended to develop global data sets on soil hydraulic properties (Dai et al., 2019a). For instance, using

a well-established global database (i.e., NCSS database), Zhang et al. (2020) proposed an ensemble of up to 13 PTFs that

allows estimates of soil water retention parameters with global coverage. However, the performance of these existing generic

ensembles could be further improved, as those studies assigned fixed weights to candidate PTFs regardless of regional soil

conditions.

Following Zhang et al. (2020), we further tested the use of AutoML-Ens to map global soil water retention parameters.

The locations of soil samples in the NCSS database cover mainly the CONUS with some data from other regions of the

world (Figure 2a), with their density distribution plotted in the USDA soil textural triangle (Figure 2b). After data quality

controls (e.g., removing some extreme soil samples with a moisture content greater than 0.6) as done by Zhang et al. (2018),

49,855 soil samples and a total of 118,599 water retention records were used with moisture content measured at matric

potentials of -0.06, -0.1, -0.33, -1, -2, or -15 bar. Since Zhang et al. (2020) have provided a comprehensive summary of the

selected PTFs (listed in Table S1), we focus mainly on comparing the estimates from AutoML-Ens with those from

individual PTFs and their three baseline ensembles (i.e., MEAN, BMA, and HME) in this work. For the predictors of

AutoML-Ens, it is noted that we do not group these PTFs according to their predictor variable requirements as in Zhang et al.

(2020) but use all potential predictors (i.e., volumetric fractions [%] of sand, silt, and clay, BD [g/cm$^3$], OC [%], and matric

potential [bar]). Additionally, the least absolute error between the predicted and observed moisture content was selected to

label the optimal PTF in the workflow. Consequently, this leads to an enclosed AutoML-assisted workflow that enables the

assignment of dynamic weights for each PTF under various environmental conditions for the final ensemble estimation.

Specifically, our goal was to achieve the following two objectives in this example: (1) to demonstrate the predictive capacity

of AutoML-Ens, especially its unique scheme of assigning dynamic weights to candidate members, and (2) to produce a set

of improved global maps of key parameters of soil water retention characteristics (i.e., field capacity and wilting points) for

global applications.

### 3.1.2 Necessity of assigning dynamic weights in ensembles

Figure 3 shows how $R^2$ and RMSE of the soil water content from the 13 PTFs and their ensembles (i.e., MEAN, BMA,

HME, and AutoML-Ens) vary across the data sets (training, testing, or overall data) and wide environmental gradients. Note

that AutoML-Ens here was defined as the leader-one ranking among all the 32 ML models involved in the AutoML

workflow, which was selected to be the stacked ensemble based on all models derived from the H2O-AutoML platform.

Results demonstrate that each PTF has distinct strengths and weaknesses in modeling underneath the data, such as the PTF

with relatively better performance or the worse one, i.e., *Wösten* PTF (Wösten et al., 1999) and *Carsel & Parrish* PTF

(Carsel and Parrish, 1988), respectively, for both the training and testing data. Further inspection indicated that the four

ensembles achieved improved predictive capabilities than any single PTF used in the analyzes, where BMA and HME

yielded better performances than MEAN. Meanwhile, AutoML-Ens was superior on the overall data with the largest positive



$R^2$ difference value of 0.075 (improved by 9% from 0.797 to 0.872) and the lowest negative RMSE difference value of -0.012 m$^3$/m$^3$ (reduced by 22% from 0.055 to 0.043 m$^3$/m$^3$) compared to the MEAN ensemble (considered as the benchmark). We further explored the variations in the $R^2$ and RMSE values of the overall 17 models under different environmental conditions (that is, different classes of USDA soil texture, matric potential, BD, and OC, as shown in Figures 3c-3j,

respectively). The general conclusions remain the same, indicating that different PTFs and their ensembles present various abilities, as expected in terms of the changing environmental gradients. More precisely, both the predictive capacities of individual PTFs and their ensembles appear to have a high sensitivity to the selected predictors. For instance, the performance of these predictions improves with increasing BD and OC values. It also suggests that those environmental factors with significant influences on model performance should not be ignored when developing models and simulating or

predicting variables.

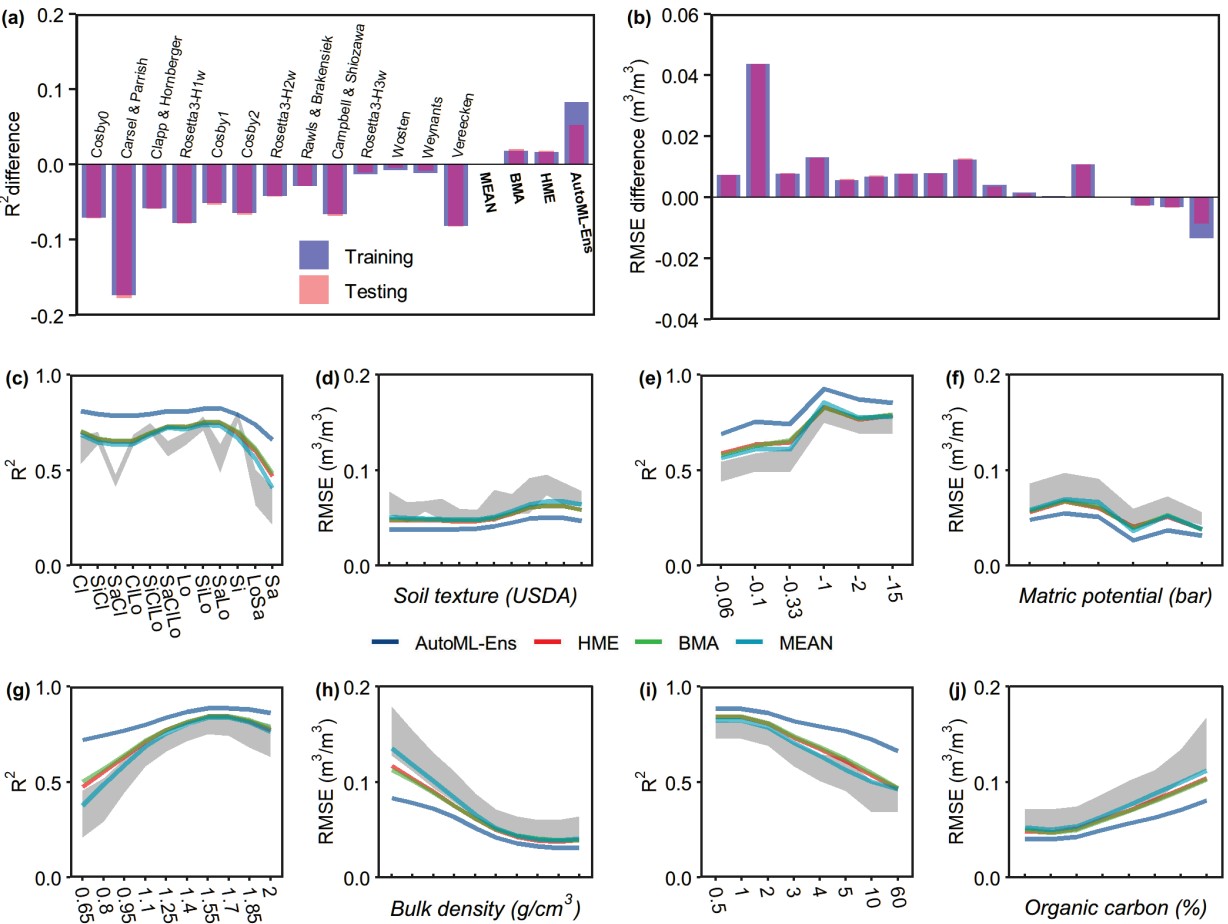

**Figure 3.** Difference in performance metrics ($R^2$ (**a**) and RMSE (**b**)) between MEAN and all 17 models, including individual PTFs and model ensembles (in bold font) for training and testing data. A positive $R^2$ or negative RMSE difference means





that the model yields a larger $R^2$ or smaller RMSE, indicating the better performance of the model than MEAN (considered
as the benchmark). $R^2$ (**c, e, g, i**) and RMSE (**d, f, h, j**) when the moisture content estimates of different ensemble approaches
were compared with observations (including all training and testing data) under various environmental conditions (6
variables, among which, the content of sand, silt, and clay was expressed together in terms of USDA soil texture classes) that
were represented by predictors for AutoML-Ens. The gray band denotes the uncertainties calculated as the mean±standard
deviation of the $R^2$ (or RMSE) values of the 13 selected PTFs.

240       In addition, ensemble PTFs are more practical due to their higher reliability and error compensation among ensemble
members. For instance, BMA weights each PTF according to its posterior model probability and offers a fixed weight for
each PTF, potentially reducing the uncertainties in individual models. However, the fixed weight assigned by these
conventional ensembles (MEAN, BMA, and HME; see Supplementary Text S1) may not fully leverage the strengths of a
PTF since it is based on the assumption that the performance of a PTF is constant under all environmental conditions. The
fact is that multiple soil factors non-linearly regulate the processes in soil water retention and further result in various
performances of individual PTFs. On the contrary, the results show clear advantages of AutoML-Ens over these
conventional ensembles on different data sets (both the training data and the testing data) and across various environmental
constraints than other ensembles and individual PTFs, highlighting its relatively better suitability for assembling multiple
PTFs for estimating soil water retention parameters.

250       Furthermore, a set of global soil water retention parameters (with a resolution of 10 km) was produced at different soil
depths (that is, 0-5 cm, 5-15 cm, 15-30 cm, 30-60 cm, 60-100 cm, and 100-200 cm) using the SoilGrids soil composition
database (Hengl et al., 2014; Hengl et al., 2017) as input for the newly proposed AutoML-Ens. Meanwhile, the ensemble
estimates based on HME were also generated for comparison (partly shown in Figure 4). Here we chose two key variables,
i.e., moisture content at -0.33 bar and -15 bar, which are commonly used to indicate field capacity and permanent wilting
point (Jury and Horton, 2004), respectively, for comparison. It can be seen in Figure 4 that despite the considerable
discrepancies in the values identified in northern high-latitude regions (> 50°N), there was a similar spatial pattern between
the ensemble estimations of HME and AutoML-Ens in most parts of the globe. Although both approaches were developed
on the basis of the same independently measured water retention data, the ensemble schemes for optimized estimations are
different. A major difference is that HME was developed for the entire data set, although a bootstrap resampling process was
adopted in optimization, in which a set of fixed weights was assigned to each PTF in all soil conditions, so that the optimized
results depended highly on the measurements. However, AutoML-Ens depicts soil conditions (predictors) as a continuum,
with the aim of finding the optimal PTF under certain environmental conditions by assigning dynamic weights for the
candidate PTFs. In other words, AutoML-Ens has learned the optimal adaptation between the predictors (environmental
constraints) and the predictions (PTFs), which allows for stronger extrapolation and increased generalization for approaching
other issues or regions. Thus, due to the limited distribution of NCSS soil samples in northern high-latitude regions, a



significant difference in the estimations from the two ensemble methods with different generalization abilities can be expected.

**(a) Water content difference at -0.33 bar: AutoML-Ens - HME**    **(b) Water content difference at -15 bar: AutoML-Ens - HME**

Water content difference (0-5 cm depth) (m³/m³)

-0.12        0        0.12        0.24        0.36

**(c) Water content at -0.33 bar: AutoML-Ens**    **(d) Water content at -15 bar: AutoML-Ens**

Water content (0-5 cm depth) (m³/m³)

0        0.1        0.2        0.3        0.4        0.5        0.6

**(e) Water content at -0.33 bar: HME**    **(f) Water content at -15 bar: HME**

Water content (0-5 cm depth) (m³/m³)

0        0.1        0.2        0.3        0.4        0.5        0.6

**Figure 4.** Global maps (with 10 km resolution) of moisture content (0-5 cm depth) with a matric potential of -0.33 bar (**a**, **c**, and **e**) and -15 bar (**b**, **d**, and **f**) delivered based on the soil composition database of SoilGrids. The first-row graphs show the differences in moisture content between the prediction of AutoML-Ens and HME. The second- and third-row graphs are ensemble predictions from AutoML-Ens and HME, respectively.




Another form of evidence on the necessity of enabling dynamic weights for an ensemble is provided in Figure 5a, which directly reflects the varying weights assigned for each PTF based on the overall data samples. As can be seen, the
weights of each PTF fluctuated dramatically with the range from approximately 0 to 1. In addition, Figures 5b and 5c illustrate global maps of PTF with the largest weight derived from AutoML-Ens among the 13 selected PTFs at a matric potential of -0.33 bar and -15 bar, respectively. As can be seen, for different soil retention parameters (e.g., water content at different matric potentials), even at the same spatial location, their PTF with the largest weight are significantly different. These again suggested that no PTF had been found to be consistently better than the other under different environmental
conditions. Therefore, if fixed weights are used in assembling these multiple PTFs for different parameters estimation, e.g., as the HME approach does, it will inevitably lead to the failure to use the advantages of different PTFs fully. However, this evaluation has some limitations because the same database (i.e., the NCSS database) was utilized to compile HME and AutoML-Ens, indicating that the two methods were not independently validated. Other evaluations and applications, for example, as input parameters to drive regional and global LSMs, need to be further conducted to indicate which product is
more accurate and reliable. Furthermore, it should be noted that regional to global scale soil parameters with a higher spatial resolution of 90 m to 1 km can also be generated through the workflow based on various data sources (e.g., recently released national gridded soil property maps of China (Liu et al., 2021)) in addition to the SoilGrids. We expect that the AutoML-Ens derived soil parameter data sets can be helpful for a variety of purposes, such as improving the performances of Earth system models.

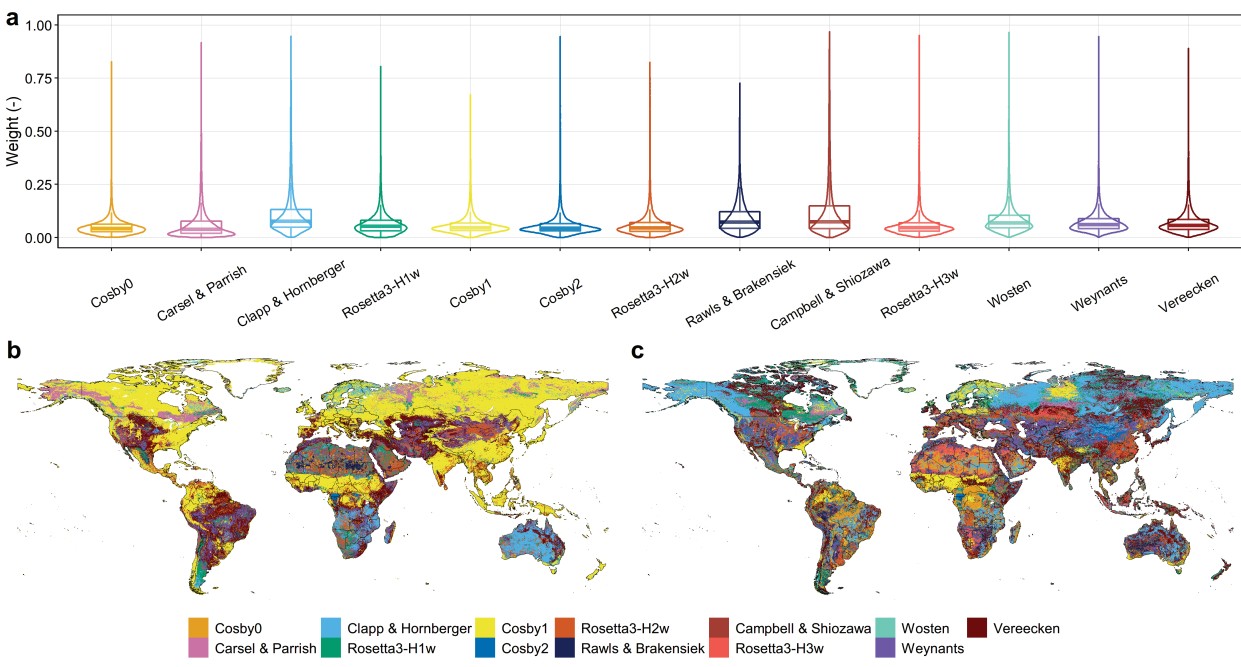




**Figure 5.** Varying weights assigned for each PTF under the overall data samples (**a**). Global maps (at 10 km resolution) of PTF with the largest weight among the 13 selected PTFs at a matric potential of -0.33 bar (**b**) and -15 bar (**c**) delivered based on the soil composition database of SoilGrids through AutoML-Ens.

In general, how to fully use the strength of individual models under certain environmental conditions is vital for making
better ensemble estimates. This example emphasizes the necessity of assigning optimal dynamic weights in ensemble approaches, which also demonstrates the great potential of AutoML-Ens to map global soil water retention-like parameters in geosciences. More specifically, for example, some observations may have already been used in calibrating the physics-based models with varying degrees, resulting in diverse performances of these models under certain environmental conditions. While the final goal of the numerous ensemble approaches is the same, that is, to obtain the final improved
estimations, they are different in ensemble strategies. It can be expected that when a physics-based model has involved more observations (i.e., more approximate to observations), the model's weight in an ensemble is relatively larger. This is especially true for conventional ensemble methods that provide fixed weights for candidate models under all conditions. However, with a varying-weight strategy under certain conditions, the advanced AutoML-Ens would not worship the model that integrates more observations nor exclude the one that may perform well under certain conditions but does not have
observation constraints. Hence, the AutoML-Ens' generalization ability is worth emphasizing.

### 3.1.3 If the classification accuracy matters?

Moreover, it is worth noting that the essence of AutoML-Ens is a kind of AutoML-assisted classifier, which also generates classification accuracy. However, improving this accuracy is not the overarching objective of AutoML-Ens. Poor accuracy may result from the uneven distribution of available data samples, their low representative ability, and inter-model
similarities and dependencies (Holtanová et al., 2019). Especially the similarities within a multi-model ensemble may result from using the same set of data samples, sharing certain components, or being based on the same hypothesis. This makes it difficult to justify the independence assumption between ensemble members, further leading to poor classification. Regarding the similarities between these 13 PTFs, it should be noted that not all PTFs were developed using independent calibration data sets, and the development legacy is not always evident. For example, data used to establish the *Rawls &*
*Brakensiek* (Rawls and Brakensiek, 1985) PTF was used by *Carsel & Parrish* and partially for the *Rosetta3* (Zhang and Schaap, 2017) PTFs. The *Vereecken* (Vereecken et al., 1989) data was used for *Weynants* (Weynants et al., 2009) PTF and also included in the database used to develop *Rosetta3* PTFs. Moreover, various ways exist by which PTFs can be grouped or distinguished, such as the predictor variable requirements (e.g., requiring the variable BD and/or OC or not) and techniques utilized (e.g., lookup table, regression, and neural networks) (Zhang et al., 2020). Furthermore, taking the derived
soil water content at -0.33 bar (0-5 cm depth) as an example, the largest weights (Figure 6a) and the difference between the largest and the second largest weights (Figure 6b) for specific PTFs are relatively small in most regions of the world. Especially, the largest weight values below 0.3 and the weight difference below 0.1 accounted for approximately 71.0% and



56.6% of the total global land area, respectively. The direct cause of this result is the similarities between these PTFs
mentioned above. However, regardless of how the selected classifier performs, the sum of the varying weights (i.e., derived
probabilities) is equal to 1 under all specified conditions. For instance, if taking the mean per class error, which indicates
misclassification of the data across the classes, as an indicator, it can be about 77% in this example. More precisely, it does
not perform very well, even for the leader model in the AutoML-Ens workflow, but has been proven to be a promising
ensemble relative to others. Therefore, efforts could be made to reduce the similarities within candidate models to obtain a
higher classification accuracy. Moreover, once a good classification accuracy is obtained among the training and testing
datasets, the linkage between the predictors and the label in the workflow will be more clearly determined, which can help
implement and/or modify these candidate models appropriately.

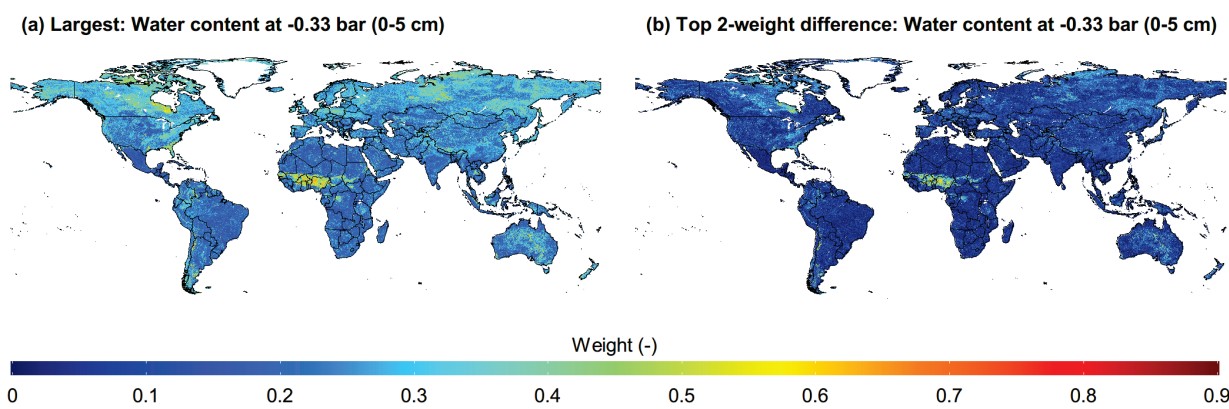

**Figure 6.** Global maps (at 10 km resolution) of the largest weight (**a**) and the top 2-weight difference values for specific PTF
(**b**) at a matric potential of -0.33 bar (0-5 cm depth) delivered based on the soil composition database of SoilGrids through
AutoML-Ens.

## 3.2 Improving remotely sensed cropland ET estimates

### 3.2.1 Related work and data acquisition

Accurate delineation of spatiotemporal variations in land ET is essential to appraise many geoscience issues, such as the
ecosystem responses to global environmental changes, but often challenging because of its highly dynamic and non-linear
response in nature (Fisher et al., 2017; Pascolini-Campbell et al., 2021; Wang and Dickinson, 2012). Given that recent
studies have shown that a multimodel ensemble can outperform individual ET models (e.g., Bai et al., 2021), the objective of
this example was to improve cropland ET estimates globally by using the AutoML-Ens framework. Following Bai et al.
(2021), observations from 47 cropland eddy covariance flux sites (listed in Table S2) covering various environmental
gradients and three continents were used (see Figures 2c-2d). Estimates from six physical ET models based on remote
sensing, namely PT-JPL, PT-DTsR, SEBS, STIC, RS-WBPM, and EVI-PM, were adopted as candidate predictions. An





overview of these six ET models is presented in Table S3. A total of 11 variables (i.e., the predictors of AutoML-Ens) jointly constraining ET based on different biophysical principles were considered, including several widely used meteorological and remote sensing factors: daily precipitation rate [mm/d], air temperature [°C], net radiation [W/m$^2$], vapor pressure deficit [hPa], wind speed [m/s], normalized vegetation index (NDVI), enhanced vegetation index (EVI), soil adjusted vegetation

index (SAVI), land surface temperature during the day (daytime LST, K), diurnal range of LST [°C], and water stress factor (0-1, a water stress factor from the RS-WBPM model (Bai et al., 2018), representing meteorological drought). After data check and filtering, a total of 83,621 records were used for ensembles and evaluations. Moreover, the least absolute errors between the daily-scale latent heat flux (LE) observations and the corresponding estimates from individual ET models were used to label the optimal physically-based ET prediction in the AutoML-Ens workflow.

**3.2.2 Advantage of an AutoML-based workflow**

Similar to the previous example results, AutoML-Ens performed much better than conventional approaches (i.e., MEAN, BMA) for assembling multiple physically-based ET models, as it yielded larger R$^2$ and smaller RMSE (Figures 7a-7b). Taken the MEAN ensemble as the benchmark, AutoML-Ens was superior on the overall data with the largest positive R$^2$ difference value of 0.15 (improved by 21.4% from 0.70 to 0.85) and the lowest negative RMSE difference value of -7.98

W/m$^2$ (reduced by 32.8% from 24.36 to 16.38 W/m$^2$). These results again suggested the importance of assigning varying weights for an ensemble because the six physically driven ET models exhibited much more complex capabilities (taking KGE as the criterion) under different environmental gradients (see Figures 7c-7m). However, some repeated evaluation results to demonstrate AutoML-Ens were omitted here. Instead, another point worth noting in this example was why the ML-based ensembles (i.e., MLP and AutoML-Ens) using almost identical datasets and procedures presented considerable

differences in terms of accuracies. As introduced by Bai et al. (2021), four different ML classifiers, namely K-nearest neighbors (KNN), MLP, random forest (RF), and support vector machine (SVM), were utilized to assemble ET models. These classifiers have different mechanisms and various schemes, thus resulting in different efficiencies among each other. On the one hand, it indicated that if other advanced ML algorithms were adopted as classifiers, MLP might not be further recognized as the best. However, on the other hand, it is too challenging to manually select the best ML classifier, which

needs the assistance of AutoML in complex pipelines. Moreover, the ranking of 32 models involved in the AutoML-Ens workflow with regard to the mean per class error was presented in Table 1. As can be seen, the best model was selected to be the stacked ensemble based on all models, followed by the stacked ensemble based on the best of family, XRT, DRF, GBM, XGBoost, and DNN, as well as their variants with different hyperparameters. This result strongly supported the need to develop ensemble approaches based on AutoML systems that inherently include ensemble learning techniques to find the

optimal combination of ML algorithms to obtain better predictive performance. Consequently, this example demonstrated and emphasized another unique feature of the proposed AutoML-Ens framework, that is, taking full advantage of the AutoML-assisted workflow. As such, AutoML-Ens, which better incorporates the capacities of diverse biophysical mechanisms and environmental variables, has the potential to improve the estimations of global cropland ET.





**Figure 7.** Difference in performance metrics ($R^2$ (**a**) and RMSE (**b**)) between MEAN and all 10 models, including six physically-based ET models and four ensembles (in bold font) for training and testing data. A positive $R^2$ or negative RMSE difference means that the model yields a larger $R^2$ or smaller RMSE, indicating the better performance of the model than MEAN (considered as the benchmark). KGE (**c-m**) when ET estimates from the 10 models were compared against observations (including all training and testing data) under various environmental conditions (11 variables) that were represented by predictors for AutoML-Ens.

**Table 1.** Ranking of the 32 models involved in the AutoML-Ens workflow with respect to the mean per class error.

| Rank | Model* | Mean per class error |
|------|--------|----------------------|




| | | |
|---|---|---|
| 1 | Stacked_Ensemble_All_Models | 0.5890107 |
| 2 | Stacked_Ensemble_Best_Of_Family | 0.5901575 |
| **3** | XRT_1 | 0.5990940 |
| 4 | DRF_1 | 0.6000693 |
| 5 | GBM_grid_1_model_1 | 0.6152126 |
| 6 | GRB_4 | 0.6156997 |
| 7 | XGBoost_grid_1_model_4 | 0.6175429 |
| 8 | XGBoost_grid_1_model_7 | 0.6182065 |
| 9 | GBM_5 | 0.6196878 |
| 10 | XGBoost_grid_1_model_9 | 0.6214154 |
| 11 | XGBoost_grid_1_model_8 | 0.6220251 |
| 12 | XGBoost_grid_1_model_1 | 0.6235140 |
| 13 | XGBoost_grid_1_model_3 | 0.6243140 |
| 14 | GBM_3 | 0.6248937 |
| 15 | XGBoost_grid_1_model_5 | 0.6252402 |
| 16 | XGBoost_grid_1_model_6 | 0.6272789 |
| 17 | GBM_grid_1_model_5 | 0.6288796 |
| 18 | XGBoost_2 | 0.6301792 |
| 19 | XGBoost_1 | 0.6313061 |
| 20 | GBM _2 | 0.6322671 |
| 21 | GBM_grid_1_model_3 | 0.6356704 |
| 22 | GBM_1 | 0.6371586 |
| 23 | XGBoost_grid_1_model_2 | 0.6444023 |
| 24 | GBM_grid_1_model_4 | 0.6470411 |
| 25 | XGBoost_3 | 0.6479244 |
| 26 | GBM_grid_1_model_2 | 0.6526127 |
| 27 | DeepLearning_grid_1_model_2 | 0.6851248 |
| 28 | DeepLearning_grid_1_model_1 | 0.6976690 |
| 29 | DeepLearning _1 | 0.7208075 |
| 30 | DeepLearning_grid_3_model_1 | 0.7247005 |
| 31 | DeepLearning_grid_2_model_1 | 0.7263856 |
| 32 | GLM_1 | 0.7417848 |

* The same ML model with different number signs indicates their variants with different hyperparameters.





### 3.2.3 Perspective on combining ML and physical modeling

Furthermore, since ML regression algorithms have been widely applied in various geoscience domains and H2O-
AutoML provides P-AutoML-Ens mentioned above based on a stacking process for assembling these algorithms, it is
interesting to address the following two more questions: (1) How does the predictive capability of AutoML-Ens compare
with those of P-AutoML-Ens? (2) What causes the differences between the performance exhibited by AutoML-Ens and P-
AutoML-Ens? To this end, we additionally built two P-AutoML-Ens workflows, taking either the observed daily scale LE or
Rn-H-G directly as labels for predicting ET as regression tasks (i.e., P-AutoML-Ens_LE and P-AutoML-Ens_Rn_H_G).
Note that Rn denotes net radiation, H and G represent sensible heat flux and ground heat flux, respectively, and in terms of
theory, 'LE = Rn - H – G' . However, due to the widely acknowledged energy balance closure problem, LE is not equal but
highly relevant to Rn-H-G for most flux observations, with an $R^2$ (RMSE) value of 0.76 (26.5 W/m$^2$) obtained in this study.
The environmental conditions (i.e., predictors) for the two workflows were the same as those for AutoML-Ens. The
comparison results are presented in Figure 8. As shown in the left part of Figures 8a-8b, first, ET estimates from no matter
the conventional ensemble methods (i.e., MEAN and BMA), the ML classifier-based ensembles with dynamic weights (i.e.,
MLP and AutoML-Ens), or P-AutoML-Ens_LE presented better performance metrics than any single physically-based ET
model, comparing against LE observations. However, it is worth noting that the performance measures of different ET
models and ensemble approaches may vary depending on the focused regions, ecosystem types, temporal scale of validation,
testing strategies, and so on. Moreover, P-AutoML-Ens_LE performed better than AutoML-Ens with slightly larger $R^2$ and
smaller RMSE, indicating that the simple regression-based P-AutoML-Ens could replace AutoML-Ens with complex physics
constraints. However, this was proven to be an illusion when we further inspected the predictive capabilities of these two
types of ensemble approaches. It was found that AutoML-Ens showed comparable performances when validated with either
the observed LE or Rn-H-G series; that is, it conserved the energy balance or followed physical constraints. In contrast,
significant discrepancies in performance metrics existed between the two P-AutoML-Ens workflows, even when the
estimations from P-AutoML-Ens_Rn_H_G were compared with the observed Rn-H-G series. This suggested that an internal
deficiency existed in these P-AutoML-Ens; that is, they cannot precisely conserve the energy budget, limiting their
extrapolation and out-of-sample generalization capacities (also discussed in Zhao et al. (2019)). Therefore, comparisons of
AutoML-Ens with P-AutoML-Ens should not be limited to a performance perspective, leading to false conclusions. Here, we
prefer to emphasize the potential of the AutoML-Ens framework, since it not only provides an effective alternative for
solving various geoscientific model ensemble problems but is well controlled by fundamental physics in geosciences.
Overall, it is worth adding here, as recent studies suggested (e.g., Jia et al., 2021; Karpatne et al., 2017; Reichstein et al.,
2019): physically-based models and ML models will not be mutually incompatible. Instead, combining ML and physical
modeling might yield a more promising but equally demanding solution.



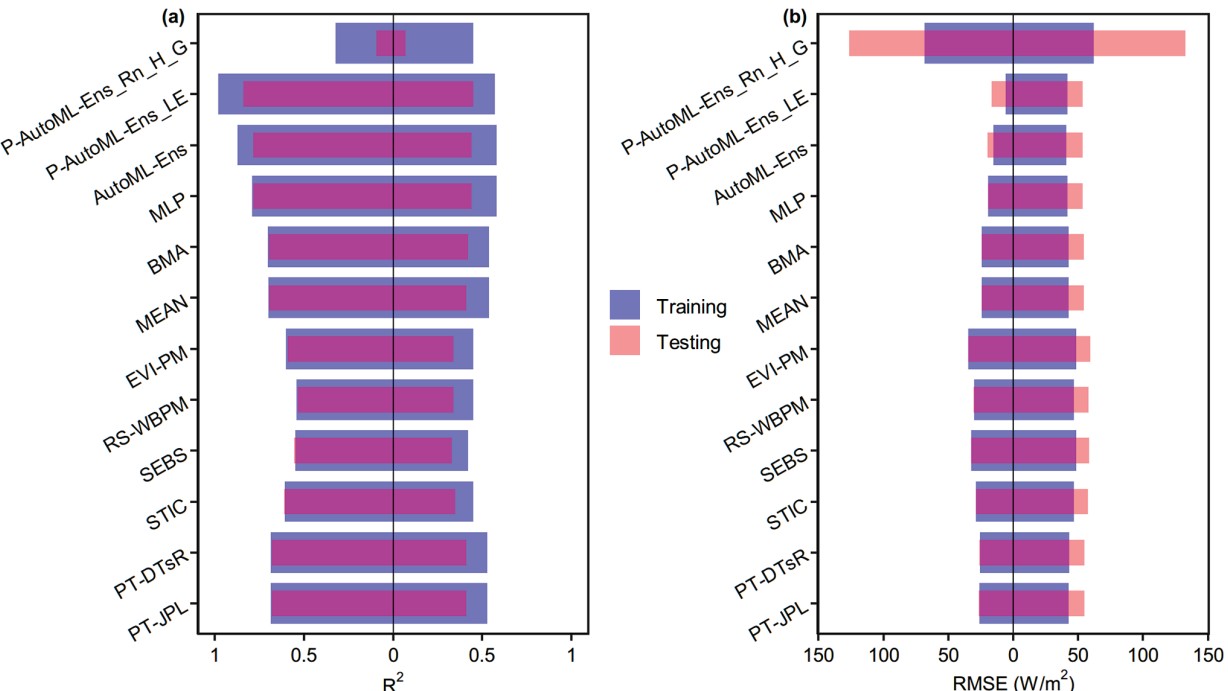

**Figure 8.** Performance metrics ($R^2$ (**a**) and RMSE (**b**)) when ET estimates from a total of 12 models, including individual six physically-based ET models and their four ensembles (i.e., MEAN, BMA, MLP, and AutoML), as well as two pure AutoML-based ensembles taking either the observed daily scale LE or Rn-H-G as labels (i.e., P-AutoML-Ens_LE or P-AutoML-Ens_Rn_H_G) in regression tasks, were compared against observations (LE (left part) and Rn-H-G (right part)) of the training and testing data.

## 4 Conclusions

The past few decades have witnessed unprecedented improvements in geoscientific modeling solutions from statistical and box models to Earth system models. However, existing models frequently utilize a few environmental factors to constrain physical processes that cannot capture fully their non-linear nature, which changes greatly across spatiotemporal domains. This is particularly true in regions with dynamic changes under the joint impact of climate change and human activities. In this study, we introduced an AutoML-Ens framework to address this issue, which could help to maximize the strengths of individual models and the ability of the unique environmental variables utilized in these models to better characterize processes. The findings lead to the following conclusions.

(1) The two illustrative applications of AutoML-Ens comprehensively demonstrated its better potential to improve estimations. Comparing to conventional ensemble approaches, AutoML-Ens produced a larger $R^2$, KGE, and smaller RMSE, for example, in estimating soil water retention parameters and cropland ET.



(2) Assigning dynamic weights to each candidate member under wide environmental conditions is essential for a better ensemble than the conventional ensemble approaches (e.g., MEAN and BMA), which usually provide fixed weights according to several statistical criteria. Specially, we proposed a novel and general strategy, i.e., mapping between ML classifier-derived probabilities and dynamic weights, in the framework. While other approaches, e.g., the known Kriging
methods, can also provide such probabilities, they can be regarded as possible extensions of the framework.

(3) Similarities within a multi-model ensemble are responsible for poor ML classification accuracy. Efforts could be devoted to reducing these similarities to obtain a higher classification accuracy. A good classification also indicates a more evident linkage between the predictors and the label in AutoML-Ens, which can, in turn, help improve these ensemble members accordingly. However, this is another critical issue that needs further exploration, and is not the overarching
objective of AutoML-Ens.

(4) Although the assignment of dynamic weights could help improve the ensembles, they are primarily based on the efficiency of ML classifiers, which require substantial human interventions for e.g., hyperparameter tuning, if done manually. Thus, taking full advantage of AutoML-assisted workflow, also one of the distinctive features of AutoML-Ens, provides a good example to guide future research in the area.

(5) Pure AutoML-based (or data-driven) ensembles may appear largely inconsistent with known physics (e.g., conservation of energy or mass), leading to an illusion of superior in model performance. Specifically, we call for the combination of data-driven approaches with physics constraints when resolving various geoscientific model ensemble issues.

**Acknowledgments**

The research is supported by the National Natural Science Foundation of China under Grant [number 42101034 and
42171036]; the China Postdoctoral Science Foundation under Grant [number 2020M680876]; and the Open Fund of State Key Laboratory of Remote Sensing Science under Grant [number OFSLRSS202110]. We thank the FLUXNET community, AmeriFlux, AsiaFlux, and the European Flux Database Cluster for providing us with eddy covariance observations. The soil database is provided by the National Cooperative Soil Survey, National Cooperative Soil Survey Soil Characterization Database.

**Code and data availability**

Processed data and source code have been made available at https://doi.org/10.6084/m9.figshare.21547134.v1. Global maps (with 10 km resolution) of field capacity and permanent wilting point at different soil depths (i.e., 0-5 cm, 5-15 cm, 15-30 cm, 30-60 cm, 60-100 cm, and 100-200 cm) derived from the hierarchical multimodel ensemble (HME) and the proposed AutoML-Ens can be downloaded online (from https://doi.org/10.6084/m9.figshare.17098487.v1).



**Author contributions**

HC was responsible for model/software curation, validation and visualization. Conceptualization and methodology development were managed by HC and YB. Writing the original manuscript was handled by HC while all authors contributed to the revision and curation of the final draft.

**Competing interests**

The authors declare that they have no conflict of interest.

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
