# Peer review of "Dynamically weighted ensemble of geoscientific models via automated machine learning-based classification"

_EGUsphere, 2022_

## Referee Comment (RC2)

Thank you for the opportunity to review this interesting study. The authors proposed the AutoML-Ens by ensembling six ML algorithms to find the best weights of predictors. Also, they considered different ensemble methods including BMA, MEAN, and so on to indicate the superior performance of the proposed method compared to these ensemble methods. In my opinion, the manuscript is suitable for publication in Geoscientific Model Development (GMD), after the authors have addressed the following comments and questions:

**Major comments:**

1- Because neural networks are one of the ML techniques and standardization is critical for this model architecture, I'm curious if the authors addressed it in the workflow.

2- According to the authors, the type of problem in this study is classification, as stated in line 159, and they utilized least absolute error to identify the ideal model (Line 118), which is used for regression problems (at least as far as I know). Maybe I misunderstood that, could you help explain this to me?

3- The authors address the accuracy of the autoML in section 3.1.3, however they don't specify the classes, and I'm curious about the proportion of classes. Is it an imbalance classification problem since the performance metric is easily skewed toward the major class? If so, how did the authors manage this situation?

4- I'm curious if the authors evaluated the predictors' correlation, as it is preferable to supply more informative information rather than a larger number of predictors for a machine learning model.

5- Generally speaking, the performance of the developed model is assessed based on benchmark. For example, multi-linear regression and logistic regression methods are used for regression and classification problems as baseline, respectively. I would like to see how well your developed model is compared to the baseline.

 **Minor comments:**

6- Figure 2 shows 47 flux sites, but the boxplots for mean annual temperature and mean annual precipitation show 44 and 42 Flux sites, respectively. Could you please clarify the differences?

7- Could you elaborate the machine learning classifier? It is hard for me to follow this term.

---

## Author Comment (AC1)

**Response to Anonymous Referee #1**

We sincerely thank the reviewer for her/his effort and the very useful comments. We have revised our manuscript *Dynamic weighted ensemble of geoscientific models via automated machine learning-based classification* and have addressed all points raised by the reviewer.

Below, we provide a point-by-point response to all the comments. Text by the reviewer is in blue and indented. Our response is in black. *New text is green, italic.* *Existing (unchanged) manuscript text is black, italic.*

> This manuscript demonstrates the merits of automatic ML (AutoML) for two geoscience use cases. In general, the paper is well written. The authors developed an ML workflow to find the best combination of models or the optimal model. They used the term ML classifier. It took me a while to understand this is different from the conventional classification problem for which the goal is to identification class labels for each sample. Instead, the goal in this work is to find the weights for combining the physics-based model ensemble.

We greatly appreciate your positive feedback. Your encouragement has significantly boosted our confidence to continue our research in this field. To highlight the innovative aspects of our research, we have made relevant modifications to the relevant sections as follows. We hope that these revisions will help to provide a clearer depiction of the concept of mapping dynamic weights to probabilities in ensembles.

*Section 2.1*

*"the ML classifier is trained to find the optimal models labeled as those that produce predictions with specific criteria (e.g., the least absolute error compared against observations for each sample of spatial/temporal predictions) under a specific environmental condition."*

*Section 3.1.1*

*"Additionally, the least absolute error between the predicted and observed moisture content was selected to label the optimal PTF for each sample in the workflow"*

> My main question is whether it is necessary to use the ensemble-based AutoML in your use cases. Can you simply use a single ML model, e.g., XGBoost, to find the model weights/probabilities? Your workflow sounds like an ensemble of ML

models for an ensemble physics models. Is this right? If so, the computational burden may be overwhelming.

In response to your main question, the first author of the manuscript, Hao Chen, provided you with a preliminary reply by way of community comment (mainly including three aspects, please refer to https://doi.org/10.5194/egusphere-2022-1326-CC1), which we hope has addressed the primary concern to some extent. Further, we would like to add some other evidence (The results are shown in a new *Table S3* in our *Supporting Information*):

*Specifically, we selected the case of assembling 13 PTFs and compared 7 different classifier configurations. These classifiers were evaluated based on their computational time and the accuracy of their ensemble predictions. Here, we utilized the H2O-AutoML platform and made use of its scenario (parameter) settings, particularly the include_algos or exclude_algos parameters (refer to the provided link: https://docs.h2o.ai/h2o/latest-stable/h2o-docs/parameters.html), to train the first 5 classifiers: 1) Original classifier (**Original_withSE**): This refers to the original classifier used in our study (consist of 6 different ML algorithms, 32 models including 2 stacked ensemble models). 2) Balanced classifier (**Balanced_withSE**): In this configuration, we enabled the "balance_classes" parameter in the original classifiers to handle the potential class imbalance issue. 3) Balanced classifier without the StackedEnsemble algorithm (**Balanced_noSE**): Here, we excluded the StackedEnsemble algorithm from the balanced classifier, meaning that no further ensemble of ML models was performed. 4) Original classifiers with only the XGBoost algorithm (**XGBoost_withSE**): Based on the original classifiers, we eliminated other algorithms except for the XGBoost and the StackedEnsemble algorithms. 5) XGBoost classifier without the StackedEnsemble algorithm (**XGBoost_noSE**): In this case, we considered only the XGBoost algorithm in the original classifier, without utilizing the StackedEnsemble algorithm. Throughout the H2O-AutoML training process, we still set the total number of models (max_models, https://docs.h2o.ai/h2o/latest-stable/h2o-docs/data-science/algo-params/max_models.html) to 30. For the 6th and 7th classifier, we opted to train the model in a Python environment using a combination of the state-of-the-art LightGBM algorithm (Ke et al., 2017) along with the efficient Optuna tool for accelerated hyperparameter optimization (Akiba et al., 2019). Note that a parameter called "n_trials", which represent the number of trials for each process in optimizing an objective function (https://optuna.readthedocs.io/en/stable/reference/generated/optuna.study.Study.html), were set to 30 and 300, respectively, thus, we obtained the other two classifiers,*

*namely **LGBM_noSE_30** and **LGBM_noSE_300**. The configuration details can be found in the code we have shared and updated recently (https://doi.org/10.6084/m9.figshare.21547134.v2).*

*__Table S3__. Computational demands and accuracy of ensemble predictions of designed machine learning classification models.*

| Classifier | Computational time (minutes) | $R^2$ (-) | RMSE ($m^3/m^3$) |
|---|---|---|---|
| Original_withSE | 84.23 | 0.8629 | 0.0444 |
| Balanced_withSE | 84.37 | 0.8654 | 0.0440 |
| Balanced_noSE | 44.54 | 0.8480 | 0.0467 |
| XGBoost_withSE | 80.28 | 0.8600 | 0.0449 |
| XGBoost_noSE | 52.40 | 0.8480 | 0.0467 |
| LGBM_noSE_30 | 12.07 | 0.8465 | 0.0472 |
| LGBM_noSE_300 | 300.33 | 0.8505 | 0.0466 |

Specifically, *__Table S3__* provides several noteworthy findings:

1) In terms of computational demands, training classifiers with an ensemble of ML models (i.e., **with_SE**) does require more time compared to **no_SE** (e.g., **Balanced_withSE** takes 47% more time than **Balanced_noSE**). However, the absolute amount of time expended remains within an acceptable range. Comparing **LGBM_noSE_30** and **LGBM_noSE_300**, even though only one model is trained, the average computational time per model still surpasses that of training 30 models using the H2O-AutoML platform.

2) Regarding accuracy, the **withSE** classifiers generally outperform the **no_SE** ones, which further supports our hypothesis that it is challenging to determine whether a ML algorithm in isolation represents the optimal solution for a given problem.

3) The issue of class imbalance has minimal impact in this example, primarily due to the modest ratio between the maximum and minimum number of classes, which is approximately 2.19 (see the new *__Table S2__*). Nonetheless, **Balanced_withSE** exhibits slight superiority over **Original_withSE**, underscoring the significance of considering class imbalance in the analysis.

**Reference**

*Akiba, T., Sano, S., Yanase, T., Ohta, T., and Koyama, M.: Optuna: A Next-generation Hyperparameter Optimization Framework, Proceedings of the 25th ACM SIGKDD*

*International Conference on Knowledge Discovery & Data Mining, Anchorage, AK, USA, 10.1145/3292500.3330701, 2019.*

*Ke, G., Meng, Q., Finley, T., Wang, T., Chen, W., Ma, W., Ye, Q., and Liu, T.-Y.: LightGBM: a highly efficient gradient boosting decision tree, Proceedings of the 31st International Conference on Neural Information Processing Systems, Long Beach, California, USA2017.*

Others minor comments:

1) Figure 3 (d)-(j). It seems all models fall outside the gray uncertainty envelope related to the 17 models. AutoML also represents an ensemble of ML models. In addition to plotting the ensemble mean from AutoML, can you develop an uncertainty envelope based on the AutoML ensemble.

Regarding Figure 3, as reply in Hao Chen's comments: we would first like to clarify that the gray bands represent the predictions of 13 PTF models, which explains why the ensemble class of models, and AutoML-Ens in particular, does not fall within this range of bands.

Here, we appreciate your suggestion, which we find to be a very good idea. However, we have made a slight adjustment to it. In addition to the 17 existing predictions, we also have included the ensemble predictions of a single ML algorithm for evaluation. Note that a particular class of ML algorithms encompasses several different variants (such as models listed in **Table 1**), and we have selected the one that demonstrates relatively good classification accuracy to represent this specific ML algorithm family. Based on this, instead update **Figure 3**, we created a new **Figure S1** in our *Supporting Information* by introducing a new band that represents the performance range (mean±standard deviation) of 6 individual ML algorithms. Note that for consistency throughout the study, we employ the classifiers derived from **Original_withSE**, as previously mentioned.

The results depicted in **Figure S1** demonstrate substantial variation in the ensemble prediction accuracy of individual ML models across specific environmental gradients, as evidenced by a wide range of $R^2$ or RMSE values. This again highlights the importance of carefully selecting the appropriate ML model for specific targets. Moreover, AutoML-Ens, as an ensemble of these ML models, exhibits prediction accuracy that, although falling within the range of ML-based ensemble accuracy, remains relatively high. This underscores the advantages of employing an ensemble ML approach in this particular case.

*"Figure S1 presents a detailed prediction comparison of 13 individual PTFs and 6 individual ML algorithms along the environmental gradients."*

[Figure]

***Figure S1***. $R^2$ (***a, c, e, g***) and RMSE (***b, d, f, h***) when the moisture content estimates of different ensemble approaches were compared with observations (including all training and testing data) under various environmental conditions (6 variables, among which, the content of sand, silt, and clay was expressed together in terms of USDA soil texture classes) that were represented by predictors for AutoML-Ens. The light gray band denotes the uncertainties calculated as the mean±standard deviation of the $R^2$ (or RMSE) values of the 13 selected PTFs. The dark gray band denotes the uncertainties of the 6 individual ML algorithms.

2) Figure 7. Both AutoML-Ens and STIC use very similar reddish color. Can you make a stronger contrast?

Modified *Figure 7*:

[Figure]

**Figure 7.** *Difference in performance metrics ($R^2$ (**a**) and RMSE (**b**)) between MEAN and all 10 models, including six physically-based ET models and four ensembles (in bold font) for training and testing data. A positive $R^2$ or negative RMSE difference means that the model yields a larger $R^2$ or smaller RMSE, indicating the better performance of the model than MEAN (considered as the benchmark). KGE (**c-m**) when ET estimates from the 10 models were compared against observations (including all training and testing data) under various environmental conditions (11 variables) that were represented by predictors for AutoML-Ens.*

Once again, we appreciate your hard work earnestly and hope that the explanations and modifications will meet with approval. If you have any other questions about this paper, please don't hesitate to let us know.

In the name of all co-authors, with kind regards.

---

## Author Comment (AC2)

**Response to Anonymous Referee #2**

We sincerely thank the reviewer for her/his effort and the very useful comments. We have revised our manuscript *Dynamic weighted ensemble of geoscientific models via automated machine learning-based classification* and have addressed all points raised by the reviewer.

Below, we provide a point-by-point response to all the comments. Text by the reviewer is in blue and indented. Our response is in black. *New text is green, italic*. *Existing (unchanged) manuscript text is black italic*.

> Thank you for the opportunity to review this interesting study. The authors proposed the AutoML-Ens by ensembling six ML algorithms to find the best weights of predictors. Also, they considered different ensemble methods including BMA, MEAN, and so on to indicate the superior performance of the proposed method compared to these ensemble methods. In my opinion, the manuscript is suitable for publication in Geoscientific Model Development (GMD), after the authors have addressed the following comments and questions.

We sincerely appreciate your comments and suggestions to improve the manuscript. However, the statement that "the AutoML-Ens by assembling six ML algorithms to find the best weights of predictors" is somehow inaccurate. In order to better address your subsequent comments, we would like to first clarify it here:

> Specifically, "*an AutoML-based training, validation, and testing workflow is conducted to help automatically find the top classifier (either a specific ML algorithm or an ensemble of a few ML algorithms based on the ensemble learning technique).*" Then, based on this classification model, we further construct predictors, which are essential input variables to develop physics-constrained models incorporated in the final ensemble. These predictors, also referred to as environmental conditions, are associated with the labels derived from physically-based model predictions that exhibit superior performance under specific environmental conditions (or for each sample). Therefore, the weights assigned in this context do not pertain to individual predictors, but instead represent the probabilities (weights) indicating the *"probability of an individual model being optimal under certain environmental conditions"*. Therefore, our focus for each sample lies not in the predicted labels produced by the ML classification model, but rather in the probabilities associated with each class of labels. These probabilities serve as the basis for determining the dynamic weights utilized in our proposed ensemble approach.

1) Because neural networks are one of the ML techniques and standardization is critical for this model architecture, I'm curious if the authors addressed it in the workflow

We extend our appreciation to the reviewer for conducting a thorough review and for raising this point. We would like to confirm that we have acknowledged the importance of standardizing variables for neural networks. However, it is noteworthy that the standardization parameter (standardize) is enabled by default in H2O-AutoML workflow, obviating the need for any specific configurations in this regard. Please refer to the following link for more details: https://docs.h2o.ai/h2o/latest-stable/h2o-docs/data-science/deep-learning.html

2) According to the authors, the type of problem in this study is classification, as stated in line 159, and they utilized least absolute error to identify the ideal model (Line 118), which is used for regression problems (at least as far as I know). Maybe I misunderstood that, could you help explain this to me?

Thank you for bringing up this important point. We would like to take this opportunity to further clarify the innovation of our study. While we acknowledge that this is a classification problem, it differs from conventional classification models in the sense that our primary focus is not solely on obtaining specific class labels. Instead, we aim to derive the probability that a prediction from various candidate members, under different environmental conditions, will be the optimal prediction for these specific conditions. This probability (i.e., weight), which is often overlooked despite being an available output of the ML classification model, plays a critical role in achieving an ensemble of model predictions at the sample scale.

Here, it is important to note that our concept of the "ideal model" does not pertain to the ML classification model itself but rather to the label (optimal prediction of physically-based models) associated with each sample. This label is utilized for data preprocessing prior to training the ML classification model. At present, we believe that the least absolute error could serve as a reasonable metric for this purpose.

Specifically, for the ML classification model, we employ the logloss metric as the loss function, as provided by H2O-AutoML for multi-classification models. Further details regarding this can be found in our code (https://doi.org/10.6084/m9.figshare.21547134.v2) or by referring to the following link: https://docs.h2o.ai/h2o/latest-stable/h2o-docs/performance-and-prediction.html

3) The authors address the accuracy of the autoML in section 3.1.3, however they don't specify the classes, and I'm curious about the proportion of classes. Is it an imbalance classification problem since the performance metric is easily skewed toward the major class? If so, how did the authors manage this situation?

Thank you for emphasizing the importance of this issue. In order to address your concern, firstly, we here provided the number of labels identified as relatively optimal (with the least absolute error) for each sample in both of our study cases (*Table S2* and added *Table S6* in our *Supporting Information*).

*Table S2*. Size of the sample labeled as individual PTFs.

| PTFs | Sample size |
|---|---|
| Cosby0 | 7,360 |
| Carsel & Parrish | 9,051 |
| Clapp & Hornberger | 12,211 |
| Rosetta3-H1w | 7,476 |
| Cosby1 | 6,884 |
| Cosby2 | 6,882 |
| Rosetta3-H2w | 6,498 |
| Rawls & Brakensiek | 10,976 |
| Campbell & Shiozawa | 14,255 |
| Rosetta3-H3w | 7,563 |
| Wösten | 11,090 |
| Weynants | 9,634 |
| Vereecken | 8,719 |

*Table S6*. Size of the sample labeled as individual ET models.

| Model name | Sample size |
|---|---|
| PT-JPL | 14,062 |
| PT-DTsR | 12,905 |
| STIC | 16,065 |
| SEBS | 12,903 |
| RS-WBPM | 16,869 |
| EVI-PM | 10,817 |

The presence of imbalanced class issues was observed in *Table S2* and *Table S6*, although they were not significant: The maximum-to-minimum ratio of class quantities in the two cases was found to be 2.19 and 1.56.

In order to assess the potential impact of not addressing this issue, a new classifier called "**Balanced_withSE**" was trained by enabling the "balance_classes"

parameter in the H2O-AutoML workflow (https://docs.h2o.ai/h2o/latest-stable/h2o-docs/data-science/algo-params/balance_classes.html). A comparison was then conducted between the "**Balanced_withSE**" classifier and the "**Original_withSE**" classifier (utilized in our study) for assembling 13 PTFs (as shown in *Table S3* in our *Supporting Information*).

The results indicated that both classifiers demonstrated very similar ensemble prediction accuracy (see the $R^2$ (0.8629 vs 0.8654) and RMSE (0.0444 vs 0.0440 $m^3/m^3$) values in *Table S3*). However, despite the small difference in our case, the "**Balanced_withSE**" classifier exhibited a slight better ensemble performance than the "**Original_withSE**" classifier. Therefore, the importance of addressing the class imbalance issue has been underscored in the main text (in *Section 2.2*) as a noteworthy key issue.

[revised manuscript text omitted]

We here still hold this opinion on these accuracies of ML classification models. Hope that the above discussion will meet with approval.

4) I'm curious if the authors evaluated the predictors' correlation, as it is preferable to supply more informative information rather than a larger number of predictors for a machine learning model.

We appreciate the reviewer's comment regarding this aspect. In order to address this key issue, we will further discuss and explain it based on our current understanding:

Indeed, when utilizing ML for predictive studies, especially in training regression models, it is crucial to conduct a thorough analysis of the correlations between predictors. This can involve performing covariance analysis, assessing variable importance, and considering the potential elimination or retention of variables based on their degree of correlation.

However, our research focuses on the ensemble of multiple physically-based models, which are formulated based on a comprehensive understanding of "*different biophysical principles*", despite their inherent limitations. These physically-based models utilize environmental variables as inputs that possess meaningful physical interpretations. Consequently, our approach aims to include a wide range of these crucial input variables, enabling ML models to utilize predictors that closely resemble those used by the physical models. This allows for more accurate comparisons between the two approaches and facilitates further exploration of the relationships between predictors and targets.

"*once a multimodel ensemble problem is defined, an extensive spectrum of* physically *meaningful predictors (i.e., environmental conditions) denoted by* $x_m$ *,*

*where $m = 1, \cdots, M$ with a single or a combination of few subsets are selected and used to develop physics-constrained models (hereafter the predictions $P_s$ where $s = 1, \cdots, S$).*"

Therefore, the selection of these predictors is depended on physics-constrained models involved in an ensemble. In our two examples, the ensemble of PTFs employed 6 environmental predictors that are essential inputs for constructing these PTFs. These predictors include matric potential, organic carbon, bulk density, and the fractions of sand, clay, and silt content. It is worth noting that there may exist simple or complex correlations among these predictors. For instance, the relationship where the sum of sand, silt, and clay fractions consistently equals 1. Similarly, in the ensemble of cropland ET models, certain key predictors (as listed in our *Supporting Information **Table S5***) such as EVI and NDVI, VPD and $T_a$ may also exhibit specific relationships. However, we would like to emphasize our intention to fully utilize the knowledge provided by physically-based models and apply it to ML approaches in an ensemble. This perspective itself deserves attention and consideration.

Moreover, we would like to highlight two recently published studies that share similarities with our approach and perspective, and may be of interest in this context: To explain (Leaf Area Index) LAI trends, Abel et al. (2023) fitted an XGBoost model using anthropogenic, climatic, topographical, and soil variables as covariates. They said that "We do not apply a variable selection procedure and instead use all available variables to parameterize the models This will ensure models with the highest possible explanatory power, and overfitting is no concern, as our aim is to explain and not to predict LAI trends". Sun et al. (2023) proposed a ML-based procedure for accelerating the spin-up of terrestrial biosphere models (TBM). For the predictors, they "consist of up to 27 variables, 20-25 variables depending on the TBM model version characterizing its driving data". It is worth noting that certain selected variables may exhibit high correlations for specific grid points on a global scale in this case.

Yet, we do hope our explanation can meet with your approval. Please let us know if you have any other comments on this issue.

We are sorry for the possible confusion regarding the ML classifier. By this point, we hope that the reviewer has gained a better understanding of the term ML classifier in our study.

The key point of our study revolves around dynamic weights, which aims to fully leverage the influence of environmental constraints on the performance of physically-based models to effectively combine the strengths of individual physically-based models under varying environmental conditions ("*i.e., weights assigned to candidate ensemble members vary depending on the spatial and temporal changes in environmental conditions and the performance capabilities of individual models under these conditions.*").

To obtain the dynamic weights, we focus on the probability predictions available within a ML classifier's outputs. While we have not conducted further tests, we speculate that certain traditional statistical methods (*e.g., the known Kriging methods*) that provide similar probabilities (weights) could also be integrated into this workflow as *possible extensions*. However, at present, we have a stronger inclination towards utilizing ML classifiers, especially when supported by extensive datasets for specific cases.

Therefore, we propose leaving this question open for readers who may further explore its significance and potential implications.

**A few words from the first author (Hao Chen):**

"When I initially considered the substitution of our frequently employed regressors with a machine learning classifier for a multi-model ensemble, I was really excited, particularly when contemplating the classifier's ability to provide not only the final predicted classes but also the probabilities associated with each class. It is a seemingly simple aspect that can be easily overlooked. While AutoML-Ens is not without its imperfections, and there remain areas requiring further in-depth exploration, I aspire to convey this potentially enlightening concept to the readers."

Once again, we appreciate your hard work earnestly and hope that the explanations and modifications will meet with approval. If you have any other questions about this paper, please don't hesitate to let us know.

In the name of all co-authors, with kind regards.

---

## Author Response (AR2)

Dear Editor,

We sincerely thank you for handling our manuscript. All the comments from you and the two reviewers are valuable and very helpful for revising and improving our paper. In the following we provide responses to the technical concerns you mentioned.

1. Certainly, the adjustment of "Dynamically" is indeed accurate and essential. Therefore, our revised title is now "Dynamically weighted ensemble of geoscientific models via automated machine learning-based classification".

2. Two additional references for the Kling-Gupta Efficiency (KGE) metric are provided:

   Gupta, H. V., Kling, H., Yilmaz, K. K., and Martinez, G. F.: Decomposition of the mean squared error and NSE performance criteria: Implications for improving hydrological modelling, Journal of Hydrology, 377(1), 80-91, https://doi.org/10.1016/j.jhydrol.2009.08.003, 2009.

   Kling, H., Fuchs, M., and Paulin, M.: Runoff conditions in the upper Danube basin under an ensemble of climate change scenarios, Journal of Hydrology, 424-425, 264-277, https://doi.org/10.1016/j.jhydrol.2012.01.011, 2012.

3. Fig.2, Fig.3, Fig. S1, and Fig.7 have all been appropriately revised in the current version of the manuscript (supplement).

4. Regarding the code and data availability:

   We sincerely apologize for any prior oversights. We have taken steps to enhance the Readme.md file by providing detailed information for each script, directory, and file involved in a new repository (https://doi.org/10.6084/m9.figshare.21547134.v3). Additionally, concerning the differences between the two versions (v1 and v2/v3), notably the absence of certain files in v2/v3, this pertains primarily to the model files generated during the training phase of our classifiers. However, it's important to acknowledge that the models constructed using different versions of the H2O-AutoML platform may not be directly transferable between versions due to ongoing updates. As a result, we suggest that readers execute the code in their individually configured environments and the primary results of the evaluation are not expected to undergo significant changes. And this shouldn't take a lot of time (under a couple hours). Moreover, this approach will facilitate readers in developing their own applications within the context of AutoML-Ens.

5. The references Ke et al., 2017 and Akiba et al., 2019 are now provided in the bibliography.

Once again, we appreciate your hard work earnestly and hope that the explanations and modifications will meet with approval. If you have any other questions about this paper, please don't hesitate to let us know.

In the name of all co-authors, with kind regards.